differential equations/mathematical physics/ applied mathematics

y-non-local Mel'nikov equation, Kadomtsev-Petviashvili hierarchy reduction method, bilinear method, soliton solutions, (semi-)rational solutions

**Author for correspondence:**
Chengfa Wu
e-mail: cfwu@szu.edu.cn

# General soliton and (semi-)rational solutions of the partial reverse space y-non-local Mel'nikov equation with non-zero boundary conditions

## Heming Fu[1], Wanshi Lu[1], Jiawei Guo[2] and Chengfa Wu[1]

[1]Institute for Advanced Study, Shenzhen University, Shenzhen 518060, People's Republic of China
[2]School of Mathematics and Statistics, University of Glasgow, Glasgow G12 8QQ, UK

CW, 0000-0003-1697-4654

General soliton and (semi-)rational solutions to the y-non-local Mel'nikov equation with non-zero boundary conditions are derived by the Kadomtsev–Petviashvili (KP) hierarchy reduction method. The solutions are expressed in $N \times N$ Gram-type determinants with an arbitrary positive integer $N$. A possible new feature of our results compared to previous studies of non-local equations using the KP reduction method is that there are two families of constraints among the parameters appearing in the solutions, which display significant discrepancies. For even $N$, one of them only generates pairs of solitons or lumps while the other one can give rise to odd numbers of solitons or lumps; the interactions between lumps and solitons are always inelastic for one family whereas the other family may lead to semi-rational solutions with elastic collisions between lumps and solitons. These differences are illustrated by a thorough study of the solution dynamics for $N = 1, 2, 3$. Besides, regularities of solutions are discussed under proper choices of parameters.

# 1. Introduction

In the past two decades, the studies on parity-time ($\mathcal{PT}$)-symmetric systems have grown significantly. A seminal work by Bender & Boettcher [1] revealed that a large class of non-Hermitian Hamiltonians exhibiting $\mathcal{PT}$-symmetry can still possess entirely real spectra. Soon afterwards, $\mathcal{PT}$-symmetry

spread out to various physical fields, such as optics [2,3], mechanical systems [4], quantum field theory [5], electric circuits [6] and many others [7]. A comprehensive review of the developments of $\mathcal{PT}$-symmetry is provided in [8].

In 2013, the concept of $\mathcal{PT}$-symmetry was introduced to integrable systems by Ablowitz & Musslimani [9]. By considering a novel non-local reduction of the Ablowitz–Kaup–Newell–Segur (AKNS) scattering problem, they proposed the non-local nonlinear Schrödinger (NLS) equation

$$\mathbf{i}q_t(x, t) = q_{xx}(x, t) + 2\sigma q(x, t)^2 q^*(-x, t), \quad \sigma = \pm 1, \tag{1.1}$$

where the asterisk $^*$ represents complex conjugation. Remarkably, this equation is $\mathcal{PT}$-symmetric as it can be viewed as a linear Schrödinger equation

$$\mathbf{i}q_t(x, t) = q_{xx}(x, t) + V(x, t)q(x, t), \tag{1.2}$$

where the self-induced potential $V(x, t) \equiv 2\sigma q(x, t)q^*(-x, t)$ admits the condition of $\mathcal{PT}$-symmetry $V(x, t) = V^*(-x, t)$. The non-locality of equation (1.1) stems from the fact that the solution's evolution at $x$ depends not only on its property at $x$, but also on its behaviour at $-x$. Subsequently, this equation has been extensively studied. Soliton solutions of equation (1.1) have been derived using various methods [10,11] and several types of rogue wave solutions of equation (1.1) were obtained via Darboux transformation [12]. With different symmetry reductions from the AKNS hierarchy and other integrable hierarchies, many new non-local equations were proposed, and some of them include the non-local complex/real sine-Gordon equation [13,14], the non-local complex/real reverse space–time modified Korteweg–de Vries (mKdV) equation [13], the non-local Davey–Stewartson (DS) equation [15–17], to name a few. The semi-discrete version [18,19] and multi-component generalizations [13,20] of the non-local NLS have been reported as well. From these studies, several distinctive features of solutions to non-local equations compared to their local counterparts were revealed, such as finite-time blow-up [9], the simultaneous existence of bright and dark solitons [21], and coexistence of solitons and kinks [22]. It should also be pointed out that non-local integrable equations may produce new physical effects and thus trigger novel physical applications. For instance, the non-local NLS (1.1) has been clarified to be related to an unconventional magnetic system [7].

Various methods of constructing exact solutions to the integrable equations have been developed, such as the Darboux transformation [23], the method of inverse scattering transformation [10], the Kadomtsev–Petviashvili (KP) hierarchy reduction method and so on [24,25]. Among them, the KP hierarchy reduction method is very powerful in deriving soliton solutions of integrable equations. This method was developed by the Kyoto school [26] and has been applied to construct soliton and breather solutions of many equations, including the NLS equation, the modified KdV equation, the Davey–Stewartson (DS) equation and the derivative Yajima–Oikawa system [27–30]. This method was also improved later to derive rogue wave and semi-rational solutions of various integrable equations [31–33] as well as their discretization [34,35]. Nevertheless, applications of this technique to non-local equations are not as successful as expected to local equations. The main obstacle is the simultaneous reductions of both the non-locality and complex conjugacy. Only very recently was this difficulty overcome by Feng *et al.* [11] in the study of soliton solutions to equation (1.1). They started with tau functions of the KP hierarchy expressed in Gram-type determinants of size 2*J*, where *J* is a positive integer. The reductions of the non-locality and complex conjugacy can be realized simultaneously by dividing the corresponding matrices into four $J \times J$ sub-matrices and imposing certain symmetry relations on the parameters in each sub-matrix. Subsequent to this, by making use of similar arguments, Rao *et al.* [36,37] obtained various solutions to the DS I equation, which contain 2*J* soliton/lump solutions and semi-rational solutions consisting of 2*J* solitons and 2*J* lumps either on the constant background or on the periodic background, where *J* is a positive integer.

Despite the successful extension of the reduction method to non-local equations, there are still some unsolved problems. On the one hand, solitons or lumps derived in both of the non-local NLS equation [11] and non-local DS I equation [36,37] always appear in pairs. On the other, the collisions between lumps and solitons that correspond to semi-rational solutions of the non-local DS I equation are inelastic. Therefore, it motivates the present work. We will solve these problems by investigating the partial reverse space y-non-local Mel'nikov equation

$$\left.\begin{array}{l} 3u_{yy} - u_{xt} - (3u^2 + u_{xx} + \kappa\Psi(x, y, t)\Psi(x, -y, t))_{xx} = 0 \\ \text{and} \quad \mathbf{i}\Psi_y = u\Psi + \Psi_{xx}, \end{array}\right\} \tag{1.3}$$

where $\kappa = \pm 1$, $u$ depicts the long wave amplitude, and $\Psi$ is the complex short wave envelope. Mel'nikov introduced the local counterpart of this equation [38,39] ($\Psi(x, -y, t)$ replaced by $\Psi^*(x, y, t)$) to model the

interaction of long waves with short wave packets. Recently, studies of the partial reverse space-time (x,t)-non-local Mel'nikov equation have been carried out in [40,41], where solutions containing even numbers of solitons or lumps were derived. Compared to them, the main contributions of this paper are listed as follows:

(a) For the reduction from the tau functions of the KP hierarchy to the bilinear equations of equation (1.3), two families of parameter relations in the $N \times N$ Gram-type determinants are found. When $N$ is even, while one of them is similar to that in [11,36,37,40], which only generates pairs of solitons or lumps, the new one can give rise to odd numbers of solitons or lumps.
(b) For even $N$, the interactions between lumps and solitons are always inelastic for the old family (similar to [36,37]) of parameter relations, whereas the new family may lead to semi-rational solutions with elastic collisions between lumps and solitons.

The rest of this paper is organized as follows. In §2, general soliton and (semi-)rational solutions of equation (1.3) are presented in theorem 2.1 and the regularity of solutions is explained in proposition 2.5. Then the proofs are provided. Sections 3 and 4 are, respectively, devoted to the discussions of soliton and (semi-)rational solutions on both constant and periodic backgrounds. We will summarize this paper in §5.

# 2. General soliton and (semi-)rational solutions of the y-non-local Mel'nikov equation

In this section, we present the general soliton and (semi-)rational solutions of equation (1.3).

## 2.1. General soliton and (semi-)rational solutions

Through the independent variable transformation

$$u = 2(\log f)_{xx}, \quad \Psi = \sqrt{2}\frac{g}{f}, \quad \Psi(x, -y, t) = \sqrt{2}\frac{h}{f}, \tag{2.1}$$

equation (1.3) can be transformed into the bilinear form

$$\left.\begin{array}{l} (D_x^4 + D_x D_t - 3D_y^2)f \cdot f = 2\kappa(cf^2 - gh) \\ (D_x^2 - iD_y)g \cdot f = 0, \end{array}\right\} \tag{2.2}$$

and

where $c$ is an arbitrary constant. Here $f$, $g$ and $h$ are functions in $x$, $y$ and $t$ that satisfy

$$f(x, y, t)g(x, -y, t) = f(x, -y, t)h(x, y, t), \tag{2.3}$$

and $D$ is Hirota's bilinear differential operator [42] defined by

$$D_x^m D_t^n f \cdot g = \left(\frac{\partial}{\partial x} - \frac{\partial}{\partial x'}\right)^m \left(\frac{\partial}{\partial t} - \frac{\partial}{\partial t'}\right)^n [f(x, t)g(x', t')]\Big|_{x'=x,t'=t}.$$

Then the soliton and (semi-)rational solutions to equation (1.3) are given as follows.

**Theorem 2.1.** *The non-local Mel'nikov equation* (1.3) *has solutions*

$$u = 2(\log f)_{xx}, \quad \Psi = \sqrt{2}\frac{g}{f}, \tag{2.4}$$

*where*

$$f = \det_{1 \le i,j \le N}\left(M_{ij}^{(0)}\right), \quad g = \det_{1 \le i,j \le N}\left(M_{ij}^{(1)}\right)$$

*with*

$$M_{ij}^{(n)} = c_i \delta_{ij} e^{-\xi_i - \eta_j}$$
$$+ \left(-\frac{p_i}{q_j}\right)^n \left(\sum_{k=0}^{n_i} a_{ik}(p_i \partial_{p_i} + \xi_i' + n)^{n_i-k} \sum_{l=0}^{n_j} b_{jl}(q_j \partial_{q_j} + \eta_j' - n)^{n_j-l}\right)\frac{1}{p_i + q_j}, \quad n = 0, 1. \tag{2.5}$$

Here $N$ is a positive integer, $\delta_{ij}$ is the Kronecker delta, the $n_i$'s are non-negative integers and

$$\left. \begin{aligned}
\xi_i &= \left( \frac{\kappa}{p_i} - 4p_i^3 \right) t + p_i x - p_i^2 y \mathbf{i} + \xi_{i0}, \\
\eta_j &= \left( \frac{\kappa}{q_j} - 4q_j^3 \right) t + q_j x + q_j^2 y \mathbf{i} + \eta_{j0}, \\
\xi_i' &= \left( -\frac{\kappa}{p_i} - 12p_i^3 \right) t + p_i x - 2p_i^2 y \mathbf{i} \\
\eta_j' &= \left( -\frac{\kappa}{q_j} - 12q_j^3 \right) t + q_j x + 2q_j^2 y \mathbf{i}.
\end{aligned} \right\} \tag{2.6}$$

and

In addition, there are two choices for parameter relations:

Case I.

(i) When $N$ is even, i.e. $N = 2J$,

$$n_{J+i} = n_i, \quad a_{J+i,k} = b_{ik}, \quad b_{J+j,l} = a_{jl}, \quad c_{J+i} = c_i \tag{2.7}$$

and

$$p_{J+i} = q_i, \quad q_{J+j} = p_j, \quad \xi_{J+i,0} = \eta_{i0}, \quad \eta_{J+j,0} = \xi_{j0}, \tag{2.8}$$

where $k = 1, 2, \ldots n_i$, $l = 1, 2, \ldots, n_j$ and $i, j = 1, 2, \ldots, J$.

(ii) When $N$ is odd, i.e. $N = 2J + 1$,

$$a_{2J+1,k} = b_{2J+1,k}, \quad p_{2J+1} = q_{2J+1}, \tag{2.9}$$

where $k = 1, 2, \ldots n_{2J+1}$. For $i, j = 1, 2, \ldots, 2J$, $p_i$, $q_j$, $c_i$, $a_{ik}$, $b_{jl}$, $\xi_{i0}$, $\eta_{j0}$ satisfy (2.7) and (2.8).

Case II. For both even and odd $N$,

$$a_{ik} = b_{ik}, \quad p_i = q_i, \tag{2.10}$$

where $k = 1, 2, \ldots n_N$ and $i = 1, 2, \ldots, N$.

**Remark 2.2.** In this paper, we focus on the dynamics of solutions with $n_i = 0$ or $1$, $i = 1, 2, \ldots, N$. Besides, without loss of generality, we may set $a_{i0} = b_{i0} = 1$, $i = 1, 2, \ldots, N$.

**Remark 2.3.** Three types of solutions (2.4) in theorem 2.1 will be discussed under different parameter restrictions: soliton solutions ($c_i \neq 0$, $n_i = 0$, $i = 1, \ldots, N$), rational solutions ($c_i = 0$, $\sum_{k=1}^N n_k \geq 1$) and semi-rational solutions ($\sum_{i=1}^N |c_i| > 0$, $\sum_{k=1}^N n_k \geq 1$).

**Remark 2.4.** For $n_1 = n_2 = \cdots = n_N = 1$, compared with Case I, the solutions (2.4) corresponding to Case II contain more free parameters $c_{[N/2]+1}, \ldots, c_{2[N/2]}$, where $\lfloor k \rfloor$ refers to the largest integer that is less than or equal to $k$. It is also noted that for even $N = 2J$, in Case I, solitons or lumps always appear in pairs, whereas Case II can give rise to odd numbers of solitons or lumps. For the dynamics of semi-rational solutions with $N = 2J$, the collisions between soliton and lump in Case I are always inelastic, while elastic collisions between them may appear for Case II. For example, if we choose

$$f = \det_{1 \leq i,j \leq 2J} \begin{pmatrix} Q_1^{(0)} & Q_2^{(0)} \\ Q_3^{(0)} & Q_4^{(0)} \end{pmatrix}, \quad g = \det_{1 \leq i,j \leq 2J} \begin{pmatrix} Q_1^{(1)} & Q_2^{(1)} \\ Q_3^{(1)} & Q_4^{(1)} \end{pmatrix}, \tag{2.11}$$

where

$$Q_1^{(n)} = \left( \left( -\frac{p_i}{q_j} \right)^n \left( \sum_{k=0}^{n_i} a_{ik}(p_i \partial_{p_i} + \xi_i' + n)^{n_i-k} \sum_{l=0}^{n_j} b_{jl}(q_j \partial_{q_j} + \eta_j' - n)^{n_j-l} \right) \frac{1}{p_i + q_j} \right)_{m \times m}$$

and

$$Q_4^{(n)} = \left( c_r \delta_{rs} \, \mathrm{e}^{-\xi_r - \eta_s} + \left( -\frac{p_r}{q_s} \right)^n \frac{1}{p_r + q_s} \right)_{(2J-m) \times (2J-m)}, \quad n = 0, 1, \ 1 \leq m < N,$$

then the corresponding semi-rational solutions may depict elastic collisions between $m$ lumps and $(2J - m)$ solitons.

**Proposition 2.5.** *The solutions (2.4) are non-singular by assuming the parameters in theorem 2.1 have further relations*

(1) $q_i = p_i^*$, $a_{ik} = b_{ik}^*$, $\xi_{i0} = \eta_{i0}^*$, $i = 1, 2, \ldots N$, $k = 1, 2, \ldots, n_N$,
(2) *all the $\Im c_i$ are positive (or negative).*

## 2.2. Proofs of theorem 2.1 and proposition 2.5

We first recall a lemma that will be needed.

**Lemma 2.6.** [31,43] *The bilinear equations in the KP hierarchy*

$$\left.\begin{array}{l} (D_{x_1}D_{x_{-1}} - 2)\tau_n \cdot \tau_n = -2\tau_{n+1}\tau_{n-1}, \\ (D_{x_1}^2 - D_{x_2})\tau_{n+1} \cdot \tau_n = 0 \\ (D_{x_1}^4 - 4D_{x_1}D_{x_3} + 3D_{x_2}^2)\tau_n \cdot \tau_n = 0, \end{array}\right\} \tag{2.12}$$

*and*

*have the Gram-type determinant solutions*

$$\tau_n = \det_{1 \le i,j \le N} (m_{ij}^{(n)}). \tag{2.13}$$

*Here, $m_{ij}^{(n)}$ are functions in $x_{-1}$, $x_1$, $x_2$ and $x_3$ defined by*

$$m_{ij}^{(n)} = c_i\delta_{ij} + A_iB_j \frac{1}{p_i + q_j}\left(-\frac{p_i}{q_j}\right)^n e^{\xi_i + \eta_j}, \tag{2.14}$$

$$\xi_i = \frac{1}{p_i}x_{-1} + p_ix_1 + p_i^2x_2 + p_i^3x_3 + \xi_{i0}, \tag{2.15}$$

$$\eta_j = \frac{1}{q_j}x_{-1} + q_jx_1 - q_j^2x_2 + q_j^3x_3 + \eta_{j0} \tag{2.16}$$

*and*

$$A_i = \sum_{k=0}^{n_i} a_{ik}(p_i\partial_{p_i})^{n_i-k}, \quad B_j = \sum_{l=0}^{m_j} b_{jl}(q_j\partial_{q_j})^{m_j-l}, \tag{2.17}$$

*where $\delta_{ij}$ is the Kronecker delta, $n_i$, $m_j$ are non-negative integers and $p_i$, $q_j$, $a_{ik}$, $b_{jl}$, $c_i$, $\xi_{i0}$, $\eta_{j0}$ are arbitrary complex constants, $i, j = 1, 2, \ldots, N$.*

Now, we consider the reductions of the bilinear equations (2.12) in the KP hierarchy to the bilinear equations (2.2), by which the soliton and (semi-)rational solutions of (1.3) can be derived. Therefore, we define

$$f = \tau_0, \quad g = \tau_1, \quad h = \tau_{-1}, \tag{2.18}$$

and take the variable transformations

$$x_1 = x, \quad x_2 = -y\mathbf{i}, \quad x_3 = -4t, \quad x_{-1} = \kappa t. \tag{2.19}$$

With these conditions, it will be shown that the functions in (2.1) satisfy equation (1.3) in theorem 2.1 as long as proper parameter constraints are imposed.

*Proof.* Proof of theorem 2.1 Denote by $m_{ij}^{(n)} = e^{\xi_i + \eta_j}M_{ij}^{(n)}$. Then with (2.19) and the operator identities

$$(p_i\partial_{p_i})p_i^n e^{\xi_i} = p_i^n e^{\xi_i}(p_i\partial_{p_i} + \xi_i' + n) \tag{2.20}$$

and

$$(q_j\partial_{q_j})q_j^{-n} e^{\eta_j} = q_j^{-n} e^{\eta_j}(q_j\partial_{q_j} + \eta_j' - n), \tag{2.21}$$

$\tau_n$ can be rewritten as

$$\tau_n = C\det_{1 \le i,j \le N} (M_{ij}^{(n)}), \tag{2.22}$$

where $C = \prod_{j=1}^N e^{\xi_j + \eta_j}$ and

$$M_{ij}^{(n)} = c_i\delta_{ij} e^{-\xi_i - \eta_j} + \left(-\frac{p_i}{q_j}\right)^n A_i'B_j' \frac{1}{p_i + q_j}, \tag{2.23}$$

with

$$A'_i = \sum_{k=0}^{n_i} a_{ik}(p_i \partial_{p_i} + \xi'_i + n)^{n_i-k}$$

and

$$B'_j = \sum_{l=0}^{m_j} b_{jl}(q_j \partial_{q_j} + \eta'_j - n)^{m_j-l}.$$

Here $\xi_i$, $\eta_j$, $\xi'_i$, $\eta'_j$ are given in (2.6).

Further, we impose parameter conditions from Case I on (2.5) for even and odd $N$.

— When $N$ is even, i.e. $N = 2J$,

$$m_i = n_i, \ \ n_{J+i} = n_i, \ \ a_{J+i,k} = b_{ik}, \ \ b_{J+j,l} = a_{jl}, \quad c_{J+i} = c_i, \tag{2.24}$$

$$p_{J+i} = q_i, \ \ q_{J+j} = p_j, \ \ \xi_{J+i,0} = \eta_{i0}, \ \ \eta_{J+j,0} = \xi_{j0}, \tag{2.25}$$

where $k = 1, 2, \ldots, n_i$, $l = 1, 2, \ldots, n_j$, $i, j = 1, 2, \ldots, J$. Thus, it follows that

$$(\xi_{J+i} + \eta_{J+j})(x, -y, t) = (\eta_i + \xi_j)(x, y, t), \tag{2.26}$$

$$\xi'_{J+i}(x, -y, t) = \eta'_i(x, y, t), \tag{2.27}$$

$$\eta'_{J+j}(x, -y, t) = \xi'_j(x, y, t). \tag{2.28}$$

Based on (2.24) and (2.26)–(2.28), we have

$$M^{(n)}_{J+i,J+j}(x, -y, t) = M^{(-n)}_{ji}(x, y, t), \tag{2.29}$$

where $i, j = 1, 2, \ldots, J$, and since $C(x, -y, t) = C(x, y, t)$, one can see that

$$\begin{aligned}
\tau_n(x, -y, t) &= C(x, -y, t)\begin{vmatrix} M^{(n)}_{ij}(x, -y, t) & M^{(n)}_{i,J+j}(x, -y, t) \\ M^{(n)}_{J+i,j}(x, -y, t) & M^{(n)}_{J+i,J+j}(x, -y, t) \end{vmatrix} \\
&= C(x, y, t)\begin{vmatrix} M^{(-n)}_{J+j,J+i}(x, y, t) & M^{(-n)}_{j,J+i}(x, y, t) \\ M^{(-n)}_{J+j,i}(x, y, t) & M^{(-n)}_{ji}(x, y, t) \end{vmatrix} \\
&= C(x, y, t)\begin{vmatrix} M^{(-n)}_{ij}(x, y, t) & M^{(-n)}_{i,J+j}(x, y, t) \\ M^{(-n)}_{J+i,j}(x, y, t) & M^{(-n)}_{J+i,J+j}(x, y, t) \end{vmatrix} \\
&= \tau_{-n}(x, y, t).
\end{aligned}$$

—When $N$ is odd, i.e. $N = 2J + 1$,

$$a_{2J+1,k} = b_{2J+1,k}, \quad p_{2J+1} = q_{2J+1}, \tag{2.30}$$

where $k = 1, 2, \ldots, n_{2J+1}$. For $i, j = 1, 2, \ldots, 2J$, $p_i, q_j, c_i, a_{ik}, b_{jl}, \xi_{i0}, \eta_{j0}$ satisfy (2.24) and (2.25). Then we have

$$(\xi_{2J+1} + \eta_{2J+1})(x, -y, t) = (\xi_{2J+1} + \eta_{2J+1})(x, y, t) \tag{2.31}$$

and

$$\xi'_{2J+1}(x, -y, t) = \eta'_{2J+1}(x, y, t). \tag{2.32}$$

Similarly, by (2.24)–(2.32), the following equations are valid:

$$M^{(n)}_{J+i,j}(x, -y, t) = M^{(-n)}_{J+j,i}(x, y, t), \tag{2.33}$$

$$M^{(n)}_{i,J+j}(x, -y, t) = M^{(-n)}_{j,J+i}(x, y, t), \tag{2.34}$$

$$M^{(n)}_{i,2J+1}(x, -y, t) = M^{(-n)}_{2J+1,J+i}(x, y, t), \tag{2.35}$$

$$M^{(n)}_{2J+1,j}(x, -y, t) = M^{(-n)}_{J+j,2J+1}(x, y, t), \tag{2.36}$$

$$M^{(n)}_{J+i,2J+1}(x, -y, t) = M^{(-n)}_{2J+1,i}(x, y, t), \tag{2.37}$$

$$M^{(n)}_{2J+1,J+j}(x, -y, t) = M^{(-n)}_{j,2J+1}(x, y, t) \tag{2.38}$$

and

$$M^{(n)}_{2J+1,2J+1}(x, -y, t) = M^{(-n)}_{2J+1,2J+1}(x, y, t). \tag{2.39}$$

From (2.29), (2.33)–(2.39) and $C(x, -y, t) = C(x, y, t)$, we have

$$\tau_n(x, -y, t) = \tau_{-n}(x, y, t).$$

Finally, by taking

$$\left.\begin{array}{l} \tau_0(x, y, t) = Cf(x, y, t), \\ \tau_1(x, y, t) = Cg(x, y, t) \\ \tau_{-1}(x, y, t) = Ch(x, y, t), \end{array}\right\} \tag{2.40}$$

and

where $C = \prod_{j=1}^{N} e^{\xi_j + \eta_j}$, we obtain

$$f(x, y, t)g(x, -y, t) = f(x, -y, t)h(x, y, t).$$

The same conclusion holds as well under the parameter constraints in Case II. Since the argument is similar to Case I, we omit the details.

This completes the proof of theorem 2.1. ∎

*Proof.* Proof of proposition 2.5. To obtain regular solutions (2.4), we need to require $f \neq 0$, i.e. $\tau_0 \neq 0$, by (2.40).

Denote by $M = (m_{ij}^{(0)})_{N \times N}$, then for any non-zero column vector $\mathbf{v} = (v_1, v_2, \ldots, v_N)^\mathrm{T}$ with its complex transpose $\bar{\mathbf{v}}$, we have

$$\begin{aligned} \bar{\mathbf{v}} M \mathbf{v} &= \sum_{i,j=1}^{N} v_i^* m_{ij}^{(0)} v_j \\ &= \sum_{i,j=1}^{N} v_i^* v_j \left( A_i B_j \frac{1}{p_i + q_j} e^{\xi_i + \eta_j} + c_i \delta_{ij} \right) \\ &= \sum_{i,j=1}^{N} v_i^* v_j \left( A_i B_j \frac{1}{p_i + q_j} e^{\xi_i + \eta_j} \right) + \sum_{i=1}^{N} c_i |v_i|^2. \end{aligned} \tag{2.41}$$

By choosing $q_i = p_i^*$, $a_{ik} = b_{ik}^*$ and $\xi_{i0} = \eta_{i0}^*$, it yields $A_i^* = B_i$ and

$$\left( \sum_{i,j=1}^{N} v_i^* v_j \left( A_i B_j \frac{1}{p_i + q_j} e^{\xi_i + \eta_j} \right) \right)^* = \sum_{i,j=1}^{N} v_i^* v_j \left( A_i B_j \frac{1}{p_i + q_j} e^{\xi_i + \eta_j} \right),$$

which implies the first term in (2.41) is real. As a consequence, if all the $\Im c_i$ are positive (or negative), then $\bar{\mathbf{v}} M \mathbf{v} \neq 0$ for any $\mathbf{v} \neq \mathbf{0}$, which gives $\tau_0 \neq 0$ and thus $f \neq 0$. The proof is completed. ∎

# 3. Dynamics of the soliton solutions

In this section, we analyse the dynamics of the soliton solutions of equation (1.3) on both constant and periodic backgrounds.

## 3.1. One-solitons and the periodic background

By taking $N = 1$ in theorem 2.1, we have the solutions

$$\Psi(x, y, t) = \sqrt{2} \frac{M_{11}^{(1)}}{M_{11}^{(0)}} = \sqrt{2} \frac{c_1 e^{-\xi_1 - \eta_1} - (p_1/q_1(p_1 + q_1))}{c_1 e^{-\xi_1 - \eta_1} + (1/(p_1 + q_1))}, \tag{3.1}$$

where

$$\xi_1 + \eta_1 = \left( \frac{1}{p_1} + \frac{1}{q_1} \right) \kappa t + (p_1 + q_1)x - (p_1^2 - q_1^2)y\mathbf{i} - 4(p_1^3 + q_1^3)t + \xi_{10} + \eta_{10}.$$

Now, we rewrite the parameters $p_1, q_1, c_1, \xi_{10}, \eta_{10}$ as

$$q_1 = p_1 = \alpha + \beta\mathbf{i}, \quad c_1 = \mu e^{v\mathbf{i}}, \quad \eta_{10} = \xi_{10} = \xi_{10R} + \xi_{10I}\mathbf{i}, \tag{3.2}$$

where $\alpha, \beta, \mu, v, \xi_{10R}, \xi_{10I}$ are real numbers and $\mu \neq 0$. Then (3.1) turns into

$$\Psi(x, y, t) = \sqrt{2} \frac{\mu e^{\omega_1 + \omega_2 \mathbf{i}} - ((\alpha - \beta\mathbf{i})2(\alpha^2 + \beta^2))}{\mu e^{\omega_1 + \omega_2 \mathbf{i}} + ((\alpha - \beta\mathbf{i})2(\alpha^2 + \beta^2))} \tag{3.3}$$

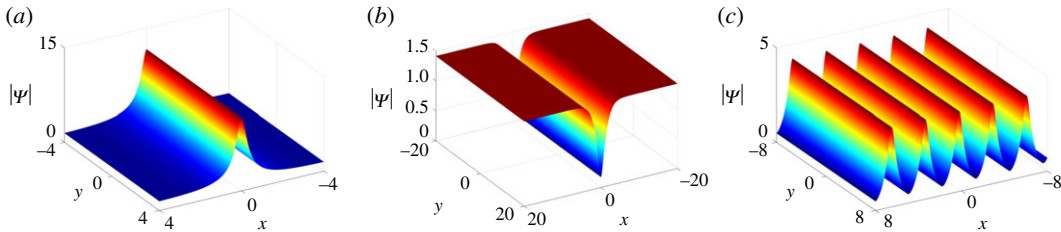

**Figure 1.** One-solitons and the periodic background of equation (1.3) at $t = 0$ with parameter values $\kappa = 1$, $\xi_{10} = 0$ and (a) $\alpha = 1/2$, $\beta = 0$, $\mu = 1$, $\nu = 29/10$, (b) $\alpha = 1/2$, $\beta = 0$, $\mu = -1$, $\nu = 29/10$, (c) $\alpha = 0$, $\beta = 1$, $\mu = 1$, $\nu = 2$.

with

$$\left. \begin{aligned} \omega_1 &= \alpha\left(\frac{-2\kappa t}{(\alpha^2 + \beta^2)} - 2x + 8(\alpha^2 - 3\beta^2)t\right) - 2\xi_{10R} \\ \omega_2 &= \beta\left(\frac{2\kappa t}{(\alpha^2 + \beta^2)} - 2x + 8(3\alpha^2 - \beta^2)t\right) - 2\xi_{10I} + \nu. \end{aligned} \right\}$$

and

Note that the solutions (3.3) are regular if its denominator is non-zero, that is

$$\mu\, e^{\omega_1} \cos\omega_2 \neq -\frac{\alpha}{2(\alpha^2 + \beta^2)} \tag{3.4}$$

or

$$\mu\, e^{\omega_1} \sin\omega_2 \neq \frac{\beta}{2(\alpha^2 + \beta^2)}. \tag{3.5}$$

Therefore, there are two cases to consider:

Case 1. When $\beta = 0$ ($p_1$ is real), the solutions (3.3) are regular for $\nu - 2\xi_{10I} \neq n\pi$. The resulting solutions

$$\Psi_s(x, y, t) = \sqrt{2}\frac{\mu\, e^{\omega_1 + (\nu - 2\xi_{10I})\mathbf{i}} - (1/2\alpha)}{\mu\, e^{\omega_1 + (\nu - 2\xi_{10I})\mathbf{i}} + (1/2\alpha)} \tag{3.6}$$

with $\omega_1 = -2\kappa t/\alpha - 2\alpha x + 8\alpha^3 t - 2\xi_{10R}$ allow one soliton independent of the variable $y$. Under different parameter conditions, we can further classify one-soliton solutions (3.6) into anti-dark soliton with $(\mu\cos(\nu - 2\xi_{10I}))/\alpha < 0$ (figure 1a), and dark soliton with $(\mu\cos(\nu - 2\xi_{10I}))/\alpha > 0$ (figure 1b).

Case 2. When $\alpha = 0$ ($p_1$ is purely imaginary), we can obtain regular solutions for $|2\mu|e^{-2\xi_{10R}} < 1/|\beta|$ satisfying condition (3.5). Then solutions (3.3) turn into

$$\Psi_p(x, y, t) = \sqrt{2}\frac{\mu\, e^{-2\xi_{10R} + \omega_2 \mathbf{i}} + (\mathbf{i}/2\beta)}{\mu\, e^{-2\xi_{10R} + \omega_2 \mathbf{i}} - (\mathbf{i}/2\beta)}, \tag{3.7}$$

where $\omega_2 = 2\kappa t/\beta - 2\beta x - 8\beta^3 t - 2\xi_{10I} + \nu$, which is independent of $y$ and periodic in both $x$ and $t$ with periods $\pi/\beta$ and $\pi\beta/(\kappa - 4\beta^4)$, respectively (figure 1c). Here the solutions (3.7) are referred to as the periodic background due to their essential roles in constructing higher-order soliton and (semi-)rational solutions on the periodic background.

## 3.2. Two- and three-soliton solutions on the constant background

*Two-soliton solutions.* To construct two-soliton solutions, we take $N = 2$ of Case I in theorem 2.1, in this case,

$$\Psi_{2s} = \sqrt{2}\frac{g}{f}, \tag{3.8}$$

where

$$f(x, y, t) = c_1^2\, e^{-(\xi_1 + \xi_2 + \eta_1 + \eta_2)} + \frac{c_1}{p_1 + q_1}(e^{-\xi_1 - \eta_1} + e^{-\xi_2 - \eta_2}) + \frac{1}{(p_1 + q_1)^2} - \frac{1}{4p_1 q_1}$$

and

$$g(x, y, t) = c_1^2\, e^{-(\xi_1 + \xi_2 + \eta_1 + \eta_2)} - \frac{c_1}{p_1 + q_1}\left(\frac{q_1}{p_1}\, e^{-\xi_1 - \eta_1} + \frac{p_1}{q_1}\, e^{-\xi_2 - \eta_2}\right) + \frac{1}{(p_1 + q_1)^2} - \frac{1}{4p_1 q_1}.$$

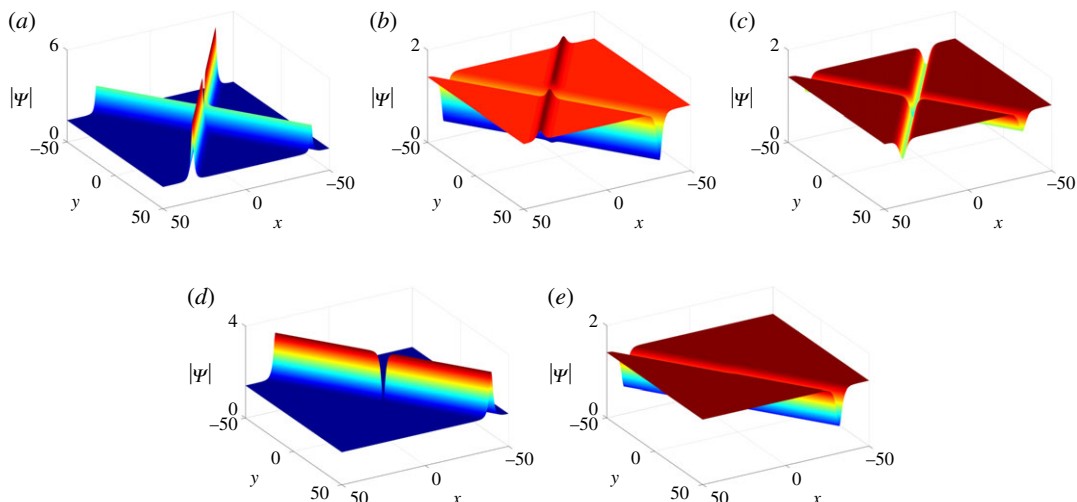

**Figure 2.** Two-soliton solutions of equation (1.3) at $t = 0$ with parameter values $\kappa = 1$, $\xi_{10} = 0$, $\rho = 1/2$, $\mu = 10$ and ($a$) $v = 5$, $\theta = 3\pi/4$, ($b$) $v = 20$, $\theta = \pi/4$, ($c$) $v = 0$, $\theta = \pi/4$; degenerate solitons: ($d$) $v = -10$, $\theta = 3\pi/4$, ($e$) $v = 10$, $\theta = \pi/4$.

**Table 1.** The classification of the two-soliton solutions under different parametric conditions.

| parametric condition I | parametric condition II | state |
|---|---|---|
| $v > -\mu \cot \theta$ | $v < \mu \cot \theta$ | dark–dark solitons |
| | $v = \mu \cot \theta$ | degenerate dark soliton |
| | $v > \mu \cot \theta$ | anti-dark–dark solitons |
| $v = -\mu \cot \theta$ | $v < \mu \cot \theta$ | degenerate dark soliton |
| | $v = \mu \cot \theta$ | constant background |
| | $v > \mu \cot \theta$ | degenerate dark soliton |
| $v < -\mu \cot \theta$ | $v < \mu \cot \theta$ | anti-dark–dark solitons |
| | $v = \mu \cot \theta$ | degenerate anti-dark soliton |
| | $v > \mu \cot \theta$ | anti-dark–anti-dark solitons |

Here $\xi_i$ and $\eta_i (i = 1, 2)$ are defined in (2.6). According to proposition 2.5, we may choose $q_1 = p_1^*$ and $\Im c_1 \neq 0$ to obtain regular solutions. Denote by $p_1 = \rho\, e^{\theta i}$, $q_1 = \rho\, e^{-\theta i}$ and $c_1 = \mu + v i$, where $0 < \theta < \pi/2$ and $\mu$, $v$ are real. Thus functions $f$, $g$ can be expressed as

$$f = (\mu + vi)^2\, e^{-(\xi_1 + \eta_1 + \xi_2 + \eta_2)} + \frac{\mu + vi}{2\rho \cos \theta} (e^{-(\xi_1 + \eta_1)} + e^{-(\xi_2 + \eta_2)}) + \frac{1}{4\rho^2 \cos^2 \theta} - \frac{1}{4\rho^2}$$

and

$$g = (\mu + vi)^2\, e^{-(\xi_1 + \eta_1 + \xi_2 + \eta_2)} - \frac{\mu + vi}{2\rho \cos \theta} (e^{-(\xi_1 + \eta_1) - 2\theta i} + e^{-(\xi_2 + \eta_2) + 2\theta i}) + \frac{1}{4\rho^2 \cos^2 \theta} - \frac{1}{4\rho^2},$$

where

$$-(\xi_1 + \eta_1) = \left( -\frac{2\kappa}{\rho} \cos \theta + 8\rho^3 \cos 3\theta \right) t - (2\rho \cos \theta) x - (2\rho^2 \sin 2\theta) y - (\xi_{10} + \eta_{10}),$$
$$-(\xi_2 + \eta_2) = \left( -\frac{2\kappa}{\rho} \cos \theta + 8\rho^3 \cos 3\theta \right) t - (2\rho \cos \theta) x + (2\rho^2 \sin 2\theta) y - (\xi_{10} + \eta_{10}).$$

Next we give the criterion of classifying the two-soliton solutions in table 1 and display five types of these two-solitons in figure 2 by choosing proper parameter values.

*Three-soliton solutions.* When $N = 3$, equation (1.3) has the solutions $\Psi_{3s} = \sqrt{2} g/f$, where

$$f = |m_{ij}^{(0)}|_{3 \times 3}, \quad g = |m_{ij}^{(1)}|_{3 \times 3},$$

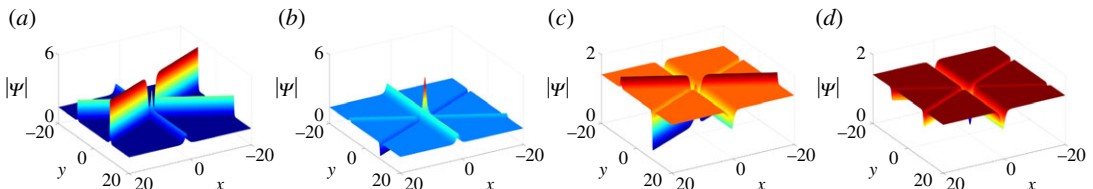

**Figure 3.** Three-soliton solutions of equation (1.3) at $t = 0$ with parameter values $\kappa = 1$, $\xi_{i0} = \eta_{i0} = 0$, $i = 1, 2, 3$, $p_2 = q_1$, $q_2 = p_1$, $q_1 = p_1^*$, $q_3 = p_3$, $p_1 = 1 + \mathbf{i}$, $c_3 = -1/2 + \mathbf{i}$ and (a) $p_3 = 1$, $c_1 = -1 + \mathbf{i}/2$, (b) $p_3 = 1$, $c_1 = 3 - 10\mathbf{i}$, (c) $p_3 = -1$, $c_1 = 3 - 10\mathbf{i}$, (d) $p_3 = -1$, $c_1 = 10 - \mathbf{i}$.

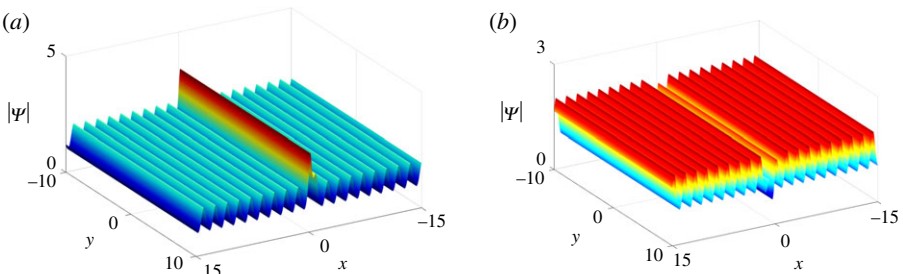

**Figure 4.** One-soliton solutions on the periodic background of equation (1.3) at $t = 0$ with parameter values $\kappa = 1$, $\xi_{10} = \xi_{20} = 0$, $p_2 = 2\mathbf{i}$, $c_1 = c_2 = 1 + \mathbf{i}$, and (a) $p_1 = -2$, (b) $p_1 = 2$.

with matrix entries

$$M_{ij}^{(n)} = c_i \delta_{ij}\, \mathrm{e}^{-\xi_i - \eta_j} + \left(-\frac{p_i}{q_j}\right)^n \frac{1}{p_i + q_j}, \quad n = 0, 1. \tag{3.9}$$

Similarly, by choosing proper parameter values (Case I in theorem 2.1), we can obtain four types of three-soliton solutions, i.e. anti-dark–anti-dark–anti-dark solitons, anti-dark–anti-dark–dark solitons, anti-dark–dark–dark solitons and dark–dark–dark solitons (figure 3).

### 3.3. One- and two-soliton solutions on the periodic background

Previously, we have derived one-solitons $\Psi_s$ (3.6), two-solitons $\Psi_{2s}$ (3.8) and the periodic background $\Psi_p$ (3.7), which naturally motivate us to obtain one- and two-soliton solutions to equation (1.3) on the periodic background.

*One-soliton solutions.* Take $N = 2$ of Case II in theorem 2.1 and we impose the parameter constraints: $q_1 = p_1$, $q_2 = p_2$. As we mentioned before, once $p_2$ is purely imaginary, one-soliton solutions on the periodic background can be obtained. By choosing proper parameter values, anti-dark soliton and dark soliton solutions on the periodic background are presented in figure 4.

*Two-soliton solutions.* Similarly, by taking $N = 3$ and $p_3$ to be purely imaginary in the basis of two-solitons $\Psi_{2s}$, five types of two-solitons can be observed on the periodic background (figure 5).

## 4. Dynamics of rational and semi-rational solutions

In this section, we will discuss the dynamical properties of rational and semi-rational solutions of equation (1.3) on both constant and periodic backgrounds.

### 4.1. Rational solutions

#### 4.1.1. One-, two- and three-lump solutions on the constant background

*One-lump solutions.* Let $N = 1$. For the convenience of subsequent discussions, we set $p_1 = q_1^*$, then equation (1.3) has one-lump solutions $\Psi_l = \sqrt{2}g/f$, where

$$f = \frac{1}{p_1 + p_1^*}\left[\frac{p_1 p_1^*}{(p_1 + p_1^*)^2} + \left(\frac{-p_1}{p_1 + p_1^*} + \xi_1 + a_{11}\right)\left(\frac{-p_1^*}{p_1 + p_1^*} + \xi_1^{\prime *} + a_{11}^*\right)\right]$$

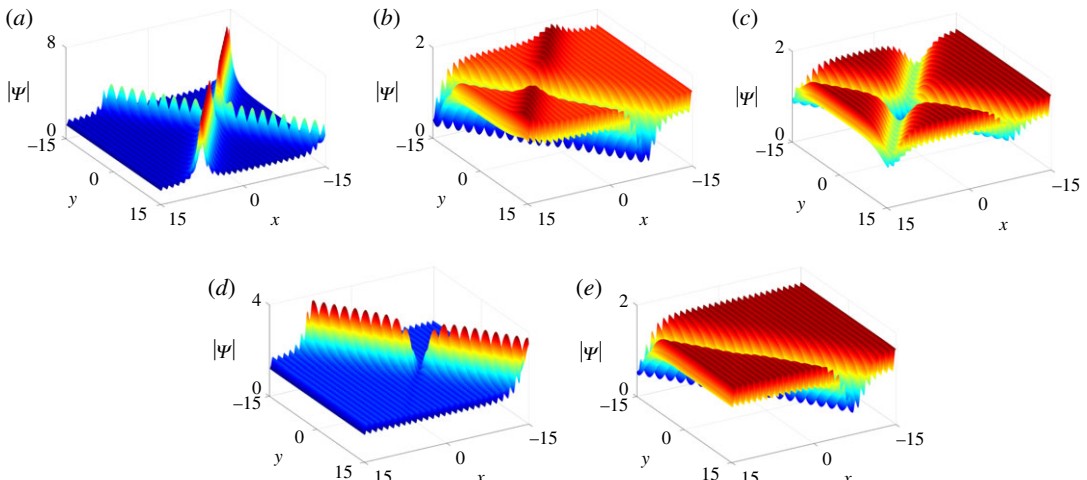

**Figure 5.** Two-soliton solutions of equation (1.3) on the periodic background at $t = 0$ with parameter values $p_3 = 3\mathbf{i}$, $c_3 = 2$, $\xi_{30} = 0$ and other parameters being the same as in figure 2.

and

$$g = -\frac{p_1}{p_1^*(p_1 + p_1^*)}\left[\frac{p_1 p_1^*}{(p_1 + p_1^*)^2} + \left(\frac{-p_1}{p_1 + p_1^*} + \xi_1' + a_{11} + 1\right)\left(\frac{-p_1^*}{p_1 + p_1^*} + \xi_1'^* + a_{11}^* - 1\right)\right].$$

Next, by taking $p_1 = \mu + \nu\mathbf{i}$ and $a_{11} = a_R + a_I\mathbf{i}$, where $\mu$, $\nu$, $a_R$ and $a_I$ are real, then we rewrite $f$ and $g$ as

$$f = \frac{1}{2\mu}(\omega\omega^* + \omega_0), \quad g = -\frac{(\mu + \nu\mathbf{i})^2}{2\mu(\mu^2 + \nu^2)}((\omega + 1)(\omega^* - 1) + \omega_0),$$

where

$$\omega = (\alpha_1 + \alpha_2\mathbf{i})x + (\beta_1 + \beta_2\mathbf{i})y + (\gamma_1 + \gamma_2\mathbf{i})t + (\lambda_1 + \lambda_2\mathbf{i}), \quad \omega_0 = \frac{\mu^2 + \nu^2}{4\mu^2},$$

$$\alpha_1 = \mu, \quad \alpha_2 = \nu, \quad \beta_1 = 4\mu\nu, \quad \beta_2 = 2\nu^2 - 2\mu^2,$$

$$\gamma_1 = \frac{-\kappa\mu}{\mu^2 + \nu^2} - 12(\mu^3 - 3\mu\nu^2), \quad \gamma_2 = \frac{\kappa\nu}{\mu^2 + \nu^2} + 12(\nu^3 - 3\mu^2\nu), \quad \lambda_1 = a_R - \frac{1}{2} \quad \text{and} \quad \lambda_2 = a_I - \frac{\nu}{2\mu}.$$

Denote by $\omega = \omega_1 + \omega_2\mathbf{i}$, where $\omega_1 = \alpha_1 x + \beta_1 y + \gamma_1 t + \lambda_1$ and $\omega_2 = \alpha_2 x + \beta_2 y + \gamma_2 t + \lambda_2$, then

$$\Psi_l = \frac{-(\mu^2 - \nu^2)(\omega_1^2 + \omega_2^2 + \omega_0 - 1) - 4\mu\nu\omega_2}{(\mu^2 + \nu^2)(\omega_1^2 + \omega_2^2 + \omega_0)} + \frac{2(-\mu\nu(\omega_1^2 + \omega_2^2 + \omega_0 - 1) - \omega_2(\mu^2 - \nu^2))\mathbf{i}}{(\mu^2 + \nu^2)(\omega_1^2 + \omega_2^2 + \omega_0)}.$$

It is easy to see that the solutions $\Psi_l$ are regular as $\mu^2 + \nu^2 \neq 0$ and $\omega_0 > 0$. Since time $t$ and $a_{11}$ represent the shifts of solutions $\Psi_l$ in time and space, respectively, we set $t = 0$ and $a_{11} = 0$ without loss of generality. After calculating the numbers of the local maximum and minimum of $|\Psi_l|$ (see appendix A), we may classify them into three patterns: bright lump ($\nu^2 \leq (1/3)\mu^2$), four-petalled lump ($(1/3)\mu^2 < \nu^2 < 3\mu^2$) and dark lump ($\nu^2 \geq 3\mu^2$).

According to (2.9), we have $p_1 = q_1$, which implies that $p_1$ and $q_1$ are real numbers, i.e. $\nu = 0$. Hence, the solutions $\Psi_l$ only generate bright lumps (figure 6).

*Two- and three-lump solutions.* We construct the two- and three-lump solutions by taking $N = 2$ and $N = 3$ of Case I in theorem 2.1, respectively, with the same matrix entries

$$M_{ij}^{(n)} = \left(-\frac{p_i}{q_j}\right)^n\left[\frac{p_i q_j}{(p_i + q_j)^2} + \left(\frac{-p_i}{p_i + q_j} + \xi_i' + n + a_{i1}\right)\left(\frac{-q_j}{p_i + q_j} + \eta_j' - n + b_{j1}\right)\right]\frac{1}{p_i + q_j}, \quad n = 0, 1.$$

According to the parameter constraints discussed in one-lump solutions, we can sort the two-lump solutions into three types, i.e. bright–bright lump solutions, four-petalled–four-petalled lump solutions and dark–dark lump solutions (see figure 7). Whereas the three-lump solutions have real parameters $p_3$ and $q_3$ leading to the three-lumps consisting of one bright lump and three types of two-lumps (figure 8).

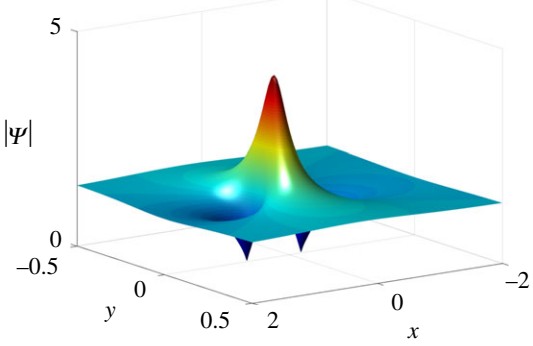

**Figure 6.** One bright lump solutions of equation (1.3) at $t = 0$ with parameter values $\kappa = 1$, $a_{11} = 0$, $\mu = 2$ and $\nu = 0$.

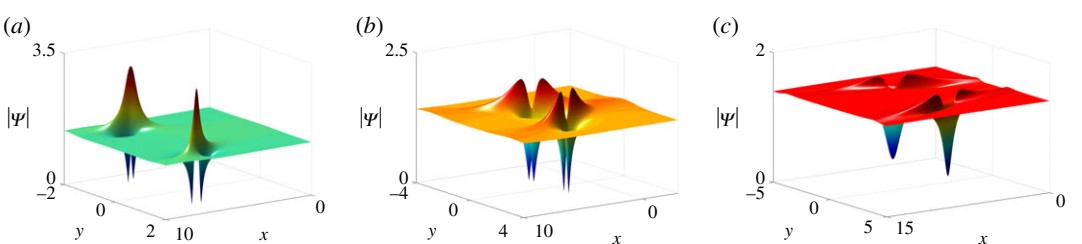

**Figure 7.** Two-lump solutions of equation (1.3) at $t = 0.1$ with parameter values $\kappa = 1$, $a_{21} = b_{11}$, $b_{21} = a_{11}$, $p_2 = q_1$, $q_2 = p_1$, $q_1 = p_1^*$, $b_{11} = a_{11}^*$, $a_{11} = 0$, and (a) $p_1 = 2 + \mathbf{i}$, (b) $p_1 = 1 + \mathbf{i}$, (c) $p_1 = 1 + 2\mathbf{i}$.

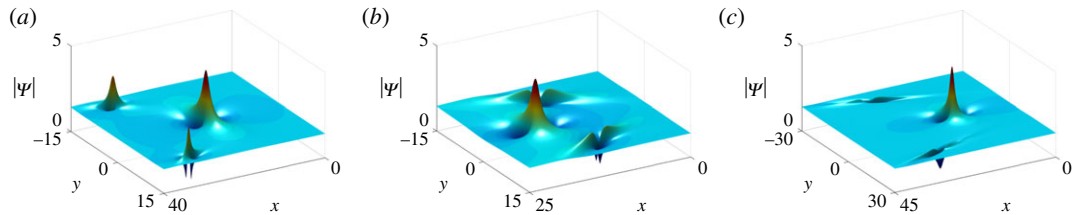

**Figure 8.** Three-lump solutions of equation (1.3) at $t = 2$ with parameter values $\kappa = 1$, $a_{21} = b_{11}$, $b_{21} = a_{11}$, $p_2 = q_1$, $q_2 = p_1$, $q_3 = p_3$, $q_1 = p_1^*$, $a_{11} = b_{11}^*$, $a_{11} = 0$, $p_3 = 1/2$, and (a) $p_1 = 1 + \mathbf{i}/2$, (b) $p_1 = 1/2 + \mathbf{i}/2$, (c) $p_1 = 1/2 + \mathbf{i}$.

### 4.1.2. One- and two-lump solutions on the periodic background

Similar to §3, we will construct one- and two-lump solutions on the periodic background.

*One-lump solutions.* We take $N = 2$ and $n_1 = 1$ of Case II to solutions of equation (1.3) in theorem 2.1, hence

$$\Psi_{lp} = \sqrt{2}\frac{g}{f}, \tag{4.1}$$

where

$$f = \begin{vmatrix} M_{11}^{(0)} & M_{12}^{(0)} \\ M_{21}^{(0)} & M_{22}^{(0)} \end{vmatrix}, \quad g = \begin{vmatrix} M_{11}^{(1)} & M_{12}^{(1)} \\ M_{21}^{(1)} & M_{22}^{(1)} \end{vmatrix}$$

with matrix entries

$$\left.\begin{aligned}
M_{11}^{(n)} &= \left(-\frac{p_1}{q_1}\right)^n \left[\frac{p_1 q_1}{(p_1 + q_1)^2} + \left(\frac{-p_1}{p_1 + q_1} + \xi_1' + n + a_{11}\right)\left(\frac{-q_1}{p_1 + q_1} + \eta_1' - n + b_{11}\right)\right]\frac{1}{p_1 + q_1}, \\
M_{12}^{(n)} &= \left(-\frac{p_1}{q_2}\right)^n \left(\frac{-p_1}{p_1 + q_2} + \xi_1' + n + a_{11}\right)\frac{1}{p_1 + q_2}, \\
M_{21}^{(n)} &= \left(-\frac{p_2}{q_1}\right)^n \left(\frac{-q_1}{p_2 + q_1} + \eta_1' - n + b_{11}\right)\frac{1}{p_2 + q_1}, \\
\text{and} \quad M_{22}^{(n)} &= c_2\, e^{-\xi_2 - \eta_2} + \left(-\frac{p_2}{q_2}\right)^n \frac{1}{p_2 + q_2}, \quad n = 0, 1.
\end{aligned}\right\} \tag{4.2}$$

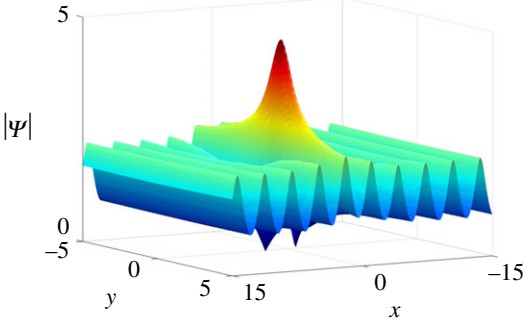

**Figure 9.** One bright lump of equation (1.3) on the periodic background at $t = 0$ with parameter values $\kappa = 1$, $a_{11} = \xi_{20} = \eta_{20} = 0$, $p_1 = 1/2$, $p_2 = \mathbf{i}$ and $c_2 = 1 + 2\mathbf{i}$.

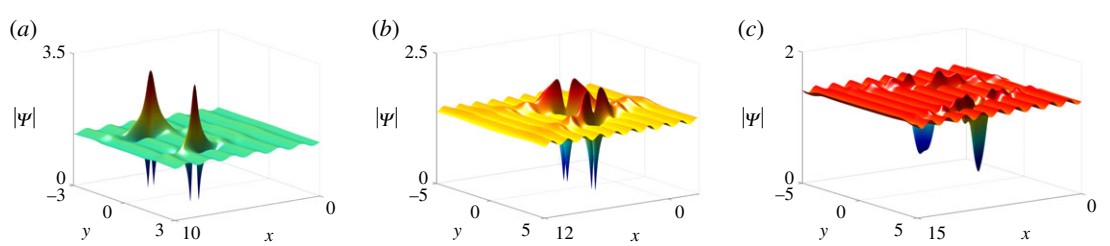

**Figure 10.** Two-lump solutions of equation (1.3) on the periodic background with parameter values $\xi_{30} = \eta_{30} = 0$, $q_3 = p_3$, $p_3 = 2\mathbf{i}$, $c_3 = 10 + 10\mathbf{i}$ and other parameters being the same as figure 7.

Based on the one-lump solutions, we set $p_2$ and $q_2$ to be purely imaginary. Naturally, we get the one bright lump on the periodic background (figure 9).

*Two-lump solutions.* Similarly, we take $N = 3$, $n_1 = n_2 = 1$ of Case I in theorem 2.1 and $p_3$, $q_3$ to be purely imaginary numbers, then three types of two-lump solutions on the periodic background can be derived (figure 10).

## 4.2. Semi-rational solutions

In this section, we continue to study the dynamics of semi-rational solutions containing combinations of solitons and lumps. First, we consider solutions on the constant background.

*One-soliton–one-lump solutions.* Interestingly, under two different parameter relations in theorem 2.1, we have found two types of collisions between soliton and lump, i.e. elastic and inelastic collisions.

Case 1. Inelastic collisions: Setting $N = 1$, the solutions are expressed as

$$\Psi_{slN1} = \sqrt{2}\frac{g}{f} = \sqrt{2}\frac{M_{11}^{(1)}}{M_{11}^{(0)}}, \tag{4.3}$$

where

$$M_{11}^{(n)} = c_1\, \mathrm{e}^{-\xi_1 - \eta_1} + \left(-\frac{p_1}{q_1}\right)^n \left[\frac{p_1 q_1}{(p_1 + q_1)^2} + \left(\frac{-p_1}{p_1 + q_1} + \xi_1' + n + a_{11}\right)\right.$$
$$\left. \times \left(\frac{-q_1}{p_1 + q_1} + \eta_1' - n + b_{11}\right)\right]\frac{1}{p_1 + q_1}, \quad n = 0, 1. \tag{4.4}$$

Here $\xi_1$, $\eta_1$, $\xi_1'$, $\eta_1'$ are defined in (2.6). To get the regular solutions by proposition 2.5, we take $q_1 = p_1^*$, $b_{11} = a_{11}^*$, and denote by $p_1 = \mu_1$ and $a_{11} = a_R$, where $\mu_1$ and $a_R$ are real. Therefore, $f$ and $g$ are rewritten as

$$f = c_1\, \mathrm{e}^{-\xi_1 - \xi_1^*} + \frac{1}{2\mu_1}(\omega\omega^* + \omega_0), \quad g = c_1\, \mathrm{e}^{-\xi_1 - \xi_1^*} - \frac{1}{2\mu_1}((\omega + 1)(\omega^* - 1) + \omega_0),$$

where

$$\omega = \mu_1 x - 2\mu_1^2 \mathbf{i} y + \left(-\frac{\kappa}{\mu_1} - 12\mu_1^3\right)t + a_R - \frac{1}{2}, \quad \omega_0 = \frac{1}{4}.$$

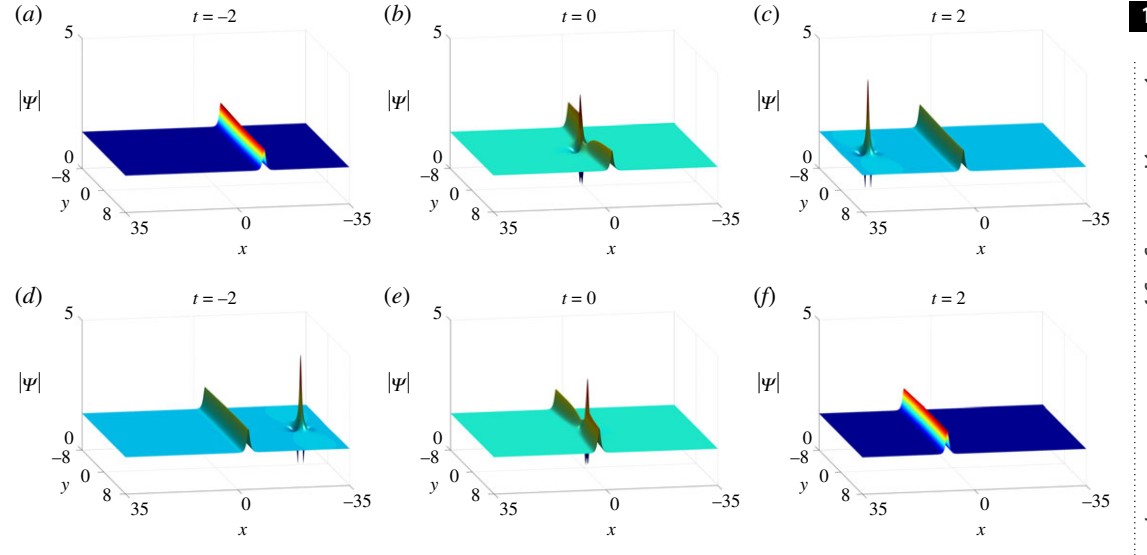

**Figure 11.** Inelastic collisions of one-soliton–one-lump solutions of equation (1.3) with parameter values $\kappa = 1$, $b_{11} = a_{11}^*$, $q_1 = p_1^*$, $a_{11} = \xi_{10} = \eta_{10} = 0$ and $(a)$–$(c)$ $p_1 = 1$, $c_1 = -1 + 2\mathbf{i}$, $(d)$–$(f)$ $p_1 = -1$, $c_1 = 1 + 2\mathbf{i}$.

Here the lump $(\omega\omega^* + \omega_0)$ moves along the line $T = \mu_1 x + (-\kappa/\mu_1 - 12\mu_1^3)t - 1/2$. Without loss of generality, we assume $a_R = 0$ and $(-\kappa/\mu_1 - 12\mu_1^3) < 0$ and investigate the asymptotic forms for the lump:

(i) Before collision $(t \to -\infty)$
   Lump $(T \approx 0,\ -\xi_1 - \xi_1^* \to -\infty)$

$$L^- = -\sqrt{2}\left(1 + \frac{\omega^* - \omega - 1}{\omega^* \omega + \omega_0}\right).$$

(ii) After collision $(t \to +\infty)$
   Lump $(T \approx 0,\ -\xi_1 - \xi_1^* \to +\infty)$

$$L^+ = \sqrt{2}.$$

From the above analysis, we can conclude that collisions between the lump and soliton are inelastic since the amplitude of the lump decreases dramatically after colliding with the soliton (figure 11).

   Case 2. Elastic collisions: By taking $q_1 = p_1 = \mu_1$, $q_2 = p_2 = \mu_2$, $b_{11} = a_{11} = a_R$ (Case II in theorem 2.1), where $\mu_1$, $\mu_2$ and $a_R$ are real, functions $f$ and $g$ can be rewritten as

$$f = \frac{c_2}{2\mu_1} e^{-\xi_2 - \xi_2^*}(\omega\omega^* + \omega_0) + \frac{(\mu_1 - \mu_2)^2(\omega\omega^* + \omega_0)}{4\mu_1\mu_2(\mu_1 + \mu_2)^2} - \frac{1}{(\mu_1 + \mu_2)^2}(\Omega^2 - (\omega + \omega^*)\Omega - \omega_0) \qquad (4.5)$$

and

$$g = -\frac{c_2}{2\mu_1} e^{-\xi_2 - \xi_2^*}(\omega\omega^* + \omega_0 + \omega^* - \omega - 1) + \frac{(\mu_1 - \mu_2)^2(\omega\omega^* + \omega_0)}{4\mu_1\mu_2(\mu_1 + \mu_2)^2} + \frac{\omega^* - \omega - 1}{4\mu_1\mu_2}$$
$$- \frac{1}{(\mu_1 + \mu_2)^2}(\Omega^2 - (\omega + \omega^*)\Omega + (\omega^* - \omega - 1) - \omega_0), \qquad (4.6)$$

where

$$\omega = \mu_1 x - 2\mu_1^2 \mathbf{i}y + \left(-\frac{\kappa}{\mu_1} - 12\mu_1^3\right)t + \left(a_R - \frac{1}{2}\right), \qquad \omega_0 = \frac{1}{4}, \qquad \Omega = \frac{\mu_1}{\mu_1 + \mu_2} - \frac{1}{2}.$$

In this case, the soliton moves along the line $(-\xi_2 - \xi_2^* = 2(4\mu_2^3 - \kappa/\mu_2)t - 2\mu_2 x - 2\xi_{20})$, while the lump $(\omega\omega^* + \omega_0)$ moves along the line $T = \mu_1 x + (-\kappa/\mu_1 - 12\mu_1^3)t + a_R - 1/2$. Without loss of generality, we assume $a_R = 0$ and $(-\kappa/\mu_1 - 12\mu_1^3) < 0$ leading to the asymptotic forms:

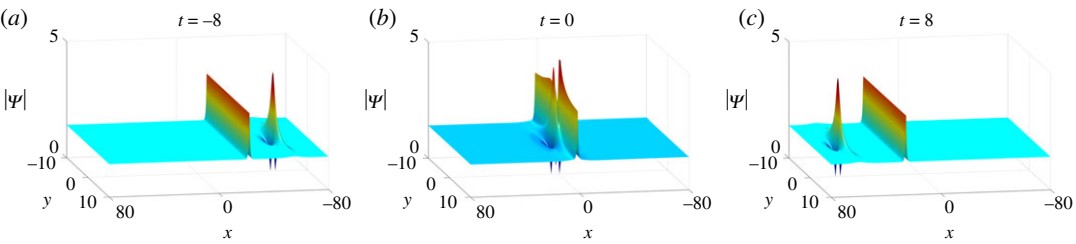

**Figure 12.** Elastic collisions of one-soliton–one-lump solutions of equation (1.3) with parameter values $\kappa = 1$, $a_{11} = 0$, $p_1 = 1/2$, $p_2 = 1$ and $c_1 = -1 + \mathbf{i}$.

(i) Before collision ($t \to -\infty$)
 Soliton ($-\xi_2 - \xi_2^* \approx 0$, $T \to +\infty$)

$$S^- \simeq -\sqrt{2}\frac{c_2\,e^{-\xi_2-\xi_2^*} - \Delta}{c_2\,e^{-\xi_2-\xi_2^*} + \Delta},$$

where $\Delta = (\mu_1 - \mu_2)^2/2\mu_2(\mu_1 + \mu_2)^2$.
 Lump ($T \approx 0$, $-\xi_2 - \xi_2^* \to -\infty$)

$$L^-(\omega, \omega^*) = \sqrt{2}\left(1 + \frac{\omega^* - \omega - 1}{\omega\omega^* + \omega_0 - \frac{4\mu_1\mu_2}{(\mu_1 - \mu_2)^2}(\Omega^2 - (\omega + \omega^*)\Omega - \omega_0)}\right)$$

$$= \sqrt{2}\left(1 + \frac{\omega^* - \omega - 1}{(\omega + \Theta)(\omega^* + \Theta) + \omega_0}\right),$$

 where $\Theta = 2\mu_1\mu_2/(\mu_1 - \mu_2)(\mu_1 + \mu_2)$.
(ii) After collision ($t \to +\infty$)
 Soliton ($-\xi_2 - \xi_2^* \approx 0$, $T \to -\infty$)

$$S^+ \simeq -\sqrt{2}\frac{c_2\,e^{-\xi_2-\xi_2^*} - \Delta}{c_2\,e^{-\xi_2-\xi_2^*} + \Delta}.$$

 Lump ($T \approx 0$, $-\xi_2 - \xi_2^* \to +\infty$)

$$L^+(\omega, \omega^*) = -\sqrt{2}\left(1 + \frac{\omega^* - \omega - 1}{\omega\omega^* + \omega_0}\right).$$

The analysis above displays that the lump and soliton undertake elastic collisions due to $|S^-| = |S^+|$ and $|L^+(\omega, \omega^*)| = |L^-(\omega + \Theta, \omega^* + \Theta)|$. Besides, the lump has experienced a phase shift $\Theta$ after the collision (figure 12).

 *One-soliton–two-lump solutions.* To obtain an one-soliton–two-lump solution, we take parameter values (Case II in theorem 2.1)

$$N = 2, \quad \kappa = 1, \quad q_1 = p_1 = \frac{1}{2}, \quad q_2 = p_2 = 1, \quad c_1 = -1/2 + \mathbf{i}, \tag{4.7}$$

$$c_2 = a_{11} = a_{21} = b_{11} = b_{21} = \xi_{10} = \xi_{20} = \eta_{10} = \eta_{20} = 0. \tag{4.8}$$

The corresponding solution $|\Psi|$ is displayed in figure 13. It is seen that one bright lump moves toward one anti-dark soliton as $t < 0$. After they interact, another new larger bright lump splits from the soliton and these two lumps move away from the soliton with different velocities.

 *Two-soliton–one-lump solutions.* The dynamics of two-soliton–one-lump solutions with parameter values (Case II in theorem 2.1)

$$N = 2, \quad \kappa = 1, \quad q_1 = p_1 = 1, \quad q_2 = p_2 = \frac{1}{2}, \quad c_1 = 1, \quad c_2 = -1 + \mathbf{i}, \tag{4.9}$$

$$a_{11} = b_{11} = \xi_{10} = \xi_{20} = \eta_{10} = \eta_{20} = 0, \tag{4.10}$$

is illustrated in figure 14. When $t < 0$ anti-dark and dark solitons move toward each other until they intersect at $t = 0$. Meanwhile, one bright lump separates from the interaction of these two solitons and gradually moves away from them.

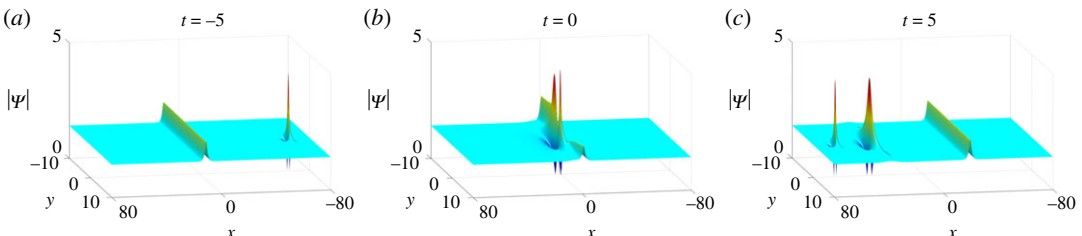

**Figure 13.** The evolution of one-soliton–two-lump solutions of equation (1.3) with parameter values in (4.7) and (4.8).

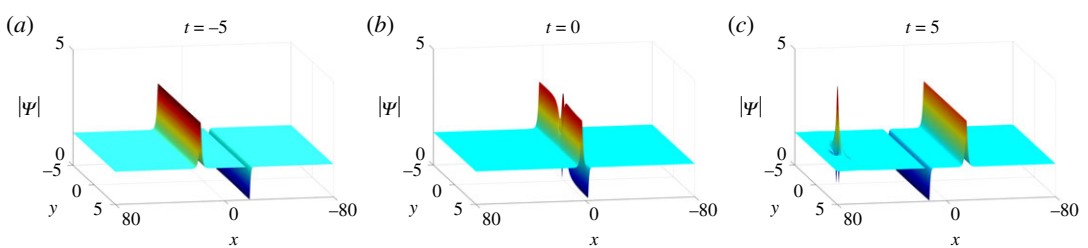

**Figure 14.** The evolution of two-soliton–one-lump solutions of equation (1.3) with parameter values in (4.9) and (4.10).

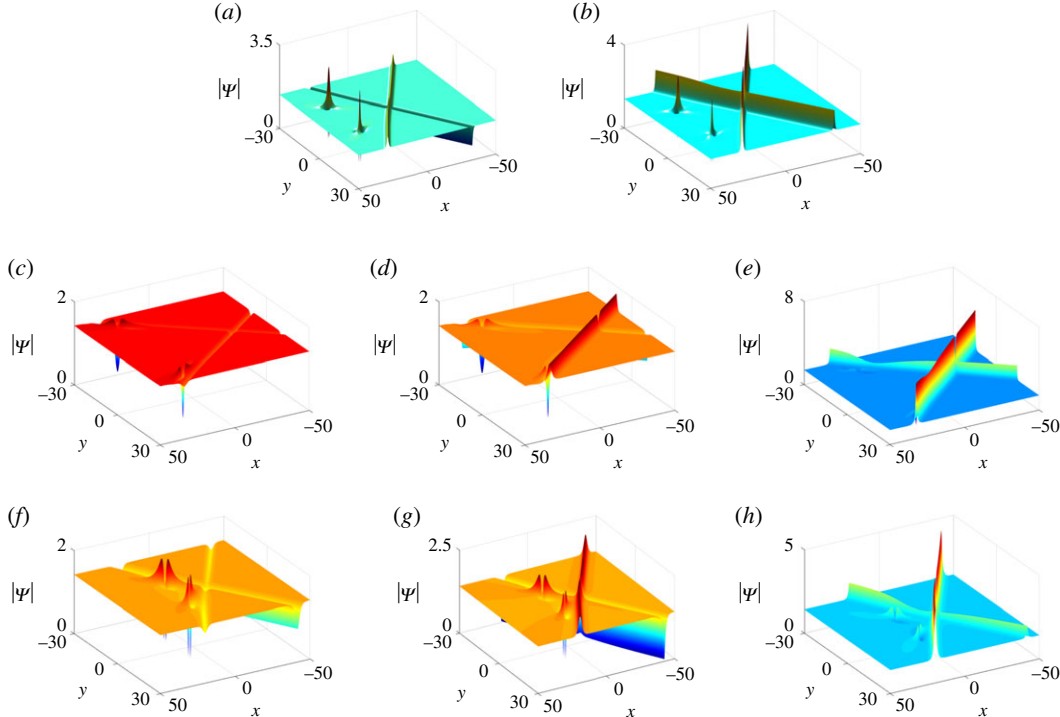

**Figure 15.** The fission of semi-rational solutions of equation (1.3). (*a*) Bright–bright lumps and anti-dark–dark solitons, (*b*) bright–bright lumps and anti-dark–anti-dark solitons, (*c*) dark–dark lumps and dark–dark solitons, (*d*) dark–dark lumps and anti-dark–dark solitons, (*e*) dark–dark lumps and anti-dark–anti-dark solitons, (*f*) four-petalled–four-petalled lumps and dark–dark solitons, (*g*) four-petalled–four-petalled lumps and anti-dark–dark solitons, (*h*) four-petalled–four-petalled lumps and anti-dark–anti-dark solitons. The corresponding parameter values are given in table 2.

*Two-soliton–two-lump solutions.* In this case, we take $N = 2$ and the parameter restrictions (Case I) listed in theorem 2.1 and proposition 2.5. Depending on the choices of parameter values, we discover nine types of semi-rational solutions consisting of two-solitons (three types) and two-lumps (three types). Since all models have similar dynamical behaviours, we just select one of them to illustrate in detail while the other eight types are shown at a specific time ($t = 2$) to demonstrate the various components (figure 15).

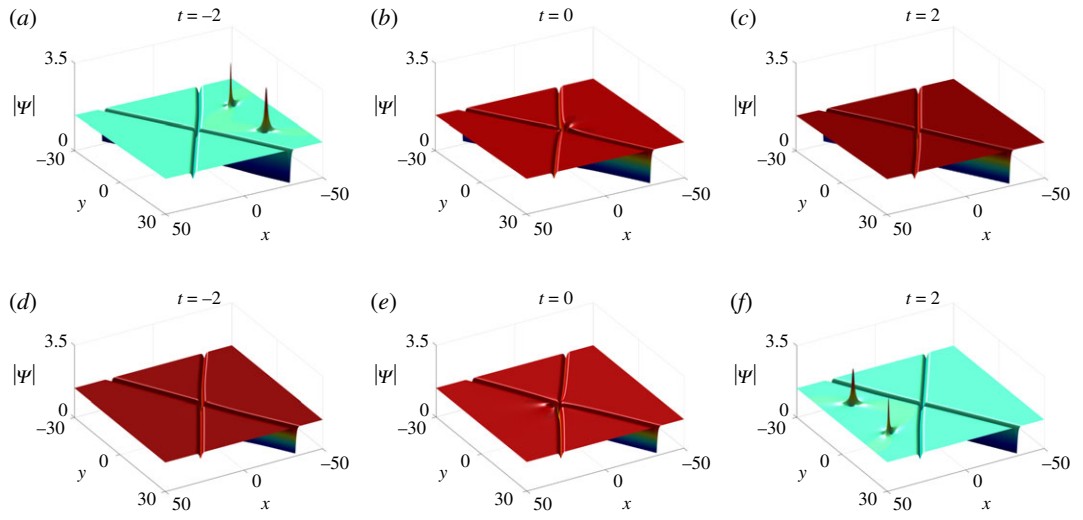

**Figure 16.** The evolution of bright–bright lump and dark–dark soliton solutions of equation (1.3) with parameter values given in table 2.

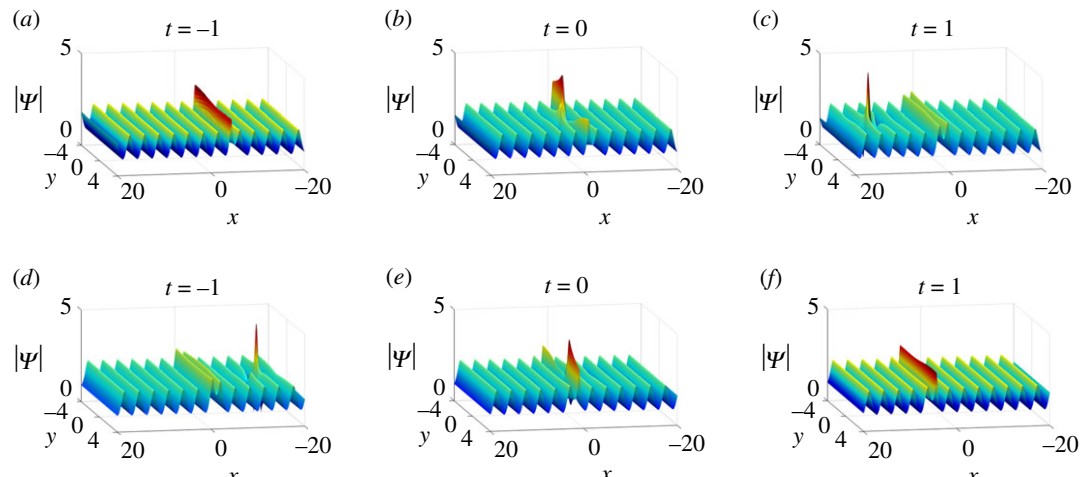

**Figure 17.** Inelastic collisions of one-soliton–one-lump solutions of equation (1.3) on the periodic background with parameter values $\kappa = 1$, $a_{11} = \xi_{10} = 0$, $p_2 = \mathbf{i}$ and (a)–(c) $p_1 = 1$, $c_1 = -1 + 2\mathbf{i}$, (d)–(f) $p_1 = -1$, $c_1 = 1 + 2\mathbf{i}$.

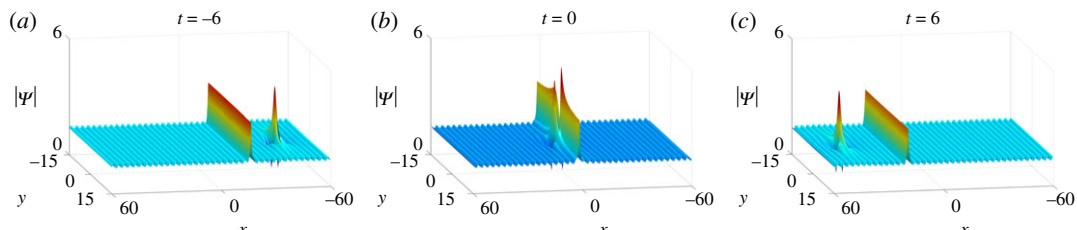

**Figure 18.** Elastic collisions of one-soliton–one-lump solutions of equation (1.3) on the periodic background with parameter values $p_1 = 1/2$, $p_2 = 1$, $p_3 = \mathbf{i}$, $c_1 = -1 + \mathbf{i}$ and $c_3 = 10 + 10\mathbf{i}$.

As shown in figure 16a–c, two bright lumps move toward two dark solitons when $t < 0$ and merge into them at $t = 0$ leaving just two solitons on the constant background. The opposite dynamical properties of these solutions are also described in figure 16d–f. When $t < 0$, two dark solitons appear and then two bright lumps gradually appear and move away from them as $t > 0$.

Finally, the dynamics of semi-rational solutions on the constant background discussed above can be extended to the periodic background (figures 17–19). The arguments are similar to those in soliton and rational solutions, and thus we omit the details.

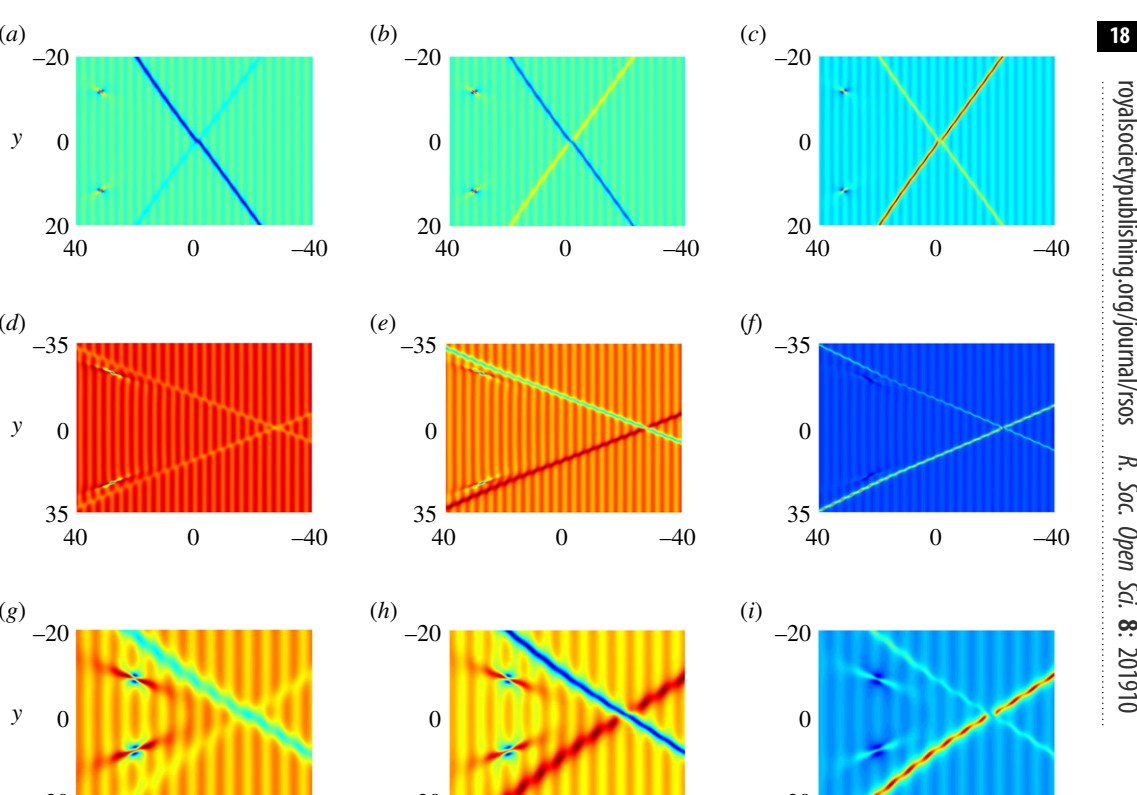

**Figure 19.** The fission of semi-rational solutions of equation (1.3) on the constant background (*a*) Bright–bright lumps and dark–dark solitons, (*b*) bright–bright lumps and anti-dark–dark solitons, (*c*) bright–bright lumps and anti-dark–anti-dark solitons, (*d*) dark–dark lumps and dark–dark solitons, (*e*) dark–dark lumps and anti-dark–dark solitons, (*f*) dark–dark lumps and anti-dark–anti-dark solitons, (*g*) four-petalled-four-petalled lumps and dark–dark solitons, (*h*) four-petalled-four-petalled lumps and anti-dark–dark solitons, (*i*) four-petalled-four-petalled lumps and anti-dark–anti-dark solitons. The corresponding parameter values are listed in table 2.

## 5. Conclusion

In this paper, we have derived general soliton and (semi-)rational solutions of equation (1.3) with non-zero boundary conditions on both constant and periodic backgrounds, by using the KP hierarchy reduction method. These solutions are expressed in terms of $N \times N$ Gram-type determinants with an arbitrary positive integer $N$, from which $N$-soliton/lump solutions can be obtained. Regularities of solutions are given in proposition 2.5 under proper choices of parameters.

Two sets of parameter relations in the Gram-type determinants are found and they demonstrate several distinctive features. The solutions (2.4) corresponding to Case II in theorem 2.1 contain more free parameters than Case I. It is also noted that for even $N = 2J$, in Case I, solitons or lumps always appear in pairs, whereas Case II can give rise to any odd number of solitons or lumps. For the dynamics of semi-rational solutions with even $N$, the collisions between soliton and lump in Case I are always inelastic, where the fission or fusion of lumps can take place, while elastic collisions between them may appear for Case II. These differences are illustrated by a comprehensive study on the dynamics of solutions for $N = 1, 2, 3$. In conclusion, compared with earlier works on the non-local Mel'nikov equation, solutions to the y-non-local Mel'nikov equation under two different cases of parameter constrains have both even and odd numbers of solitons/lumps and richer dynamical behaviours.

To the best of our knowledge, the results obtained in this paper are entirely new and provide a further extension of the KP hierarchy reduction method to non-local equations. Finally, the physical implications of our results await future efforts of researchers.

Data accessibility. This article has no additional data.
Authors' contributions. H.F. carried out the derivation of all solutions to the non-local Mel'nikov equation obtained in this paper and drafted the manuscript. W.L. and J.G. carried out the analysis of solution dynamics and the drawing of

figures and critically revised the manuscript. C.W. conceived of the study, designed the study, coordinated the study, and helped draft the manuscript. All authors gave final approval for publication and agree to be held accountable for the work performed therein.

Competing interests. We declare we have no competing interests.

Funding. This work was supported by the National Natural Science Foundation of China (grant nos. 11701382 and 11971288) and Guangdong Basic and Applied Basic Research Foundation (grant no. 2021A1515010054).

# Appendix A

In this appendix, we present the classification of first-order rational solutions. First, we can calculate the critical points of $|\Psi_l|^2$ defined in §4.1.1, which are

$$K_1 : (x_1, y_1) = \left( \frac{1}{2\mu}, 0 \right),$$

$$K_2 : (x_2, y_2) = \left( \frac{2v\sqrt{3v^2 - \mu^2} + (\mu^2 + v^2)}{2\mu(\mu^2 + v^2)}, -\frac{\sqrt{3v^2 - \mu^2}}{4\mu(\mu^2 + v^2)} \right), \quad K_3 : (x_3, y_3) = (-x_2, -y_2),$$

$$K_4 : (x_4, y_4) = \left( \frac{(v^2 - \mu^2)\sqrt{3\mu^2 - v^2} + \mu(\mu^2 + v^2)}{2\mu^2(\mu^2 + v^2)}, -\frac{v\sqrt{3\mu^2 - v^2}}{4\mu^2(\mu^2 + v^2)} \right), \quad K_5 : (x_5, y_5) = (-x_4, -y_4).$$

Let $H(x, y) = \partial^2 |\Psi_l|^2 / \partial x^2 (\partial^2 |\Psi_l|^2 / \partial y^2) - (\partial^2 |\Psi_l|^2 / \partial x \partial y)^2$, then it follows that

$$(|\Psi_l|^2)_{xx}(x_1, y_1) = -\frac{192\mu^4(\mu^2 - v^2)}{(\mu^2 + v^2)^2}, \quad H(x_1, y_1) = \frac{16384\mu^{10}(3\mu^2 - v^2)(\mu^2 - 3v^2)}{(\mu^2 + v^2)^4},$$

$$(|\Psi_l|^2)_{xx}(x_2, y_2) = (|\Psi_l|^2)_{xx}(x_3, y_3) = -\frac{6\mu^4(\mu^2 + v^2)}{v^4},$$

$$H(x_2, y_2) = H(x_3, y_3) = -\frac{64\mu^{10}(\mu^2 - 3v^2)(\mu^2 + v^2)^2}{v^{10}},$$

$$(|\Psi_l|^2)_{xx}(x_4, y_4) = (|\Psi_l|^2)_{xx}(x_5, y_5) = 6(\mu^2 + v^2),$$

$$H(x_4, y_4) = H(x_5, y_5) = 64(3\mu^2 - v^2)(\mu^2 + v^2)^2.$$

By applying the second derivative test for local extrema, first-order rational solutions fall into three types:

(I) Bright lump: when $v^2 \leq (1/3)\mu^2$, $|\Psi_l|$ has one local maximum at $K_1$ and two local minima at $K_4$ and $K_5$;

(II) Four-petalled lump: when $(1/3)\mu^2 < v^2 < 3\mu^2$, $|\Psi_l|$ has two local maxima at $K_2$ and $K_3$ and two local minima at $K_4$ and $K_5$;

(III) Dark lump: when $v^2 \geq 3\mu^2$, $|\Psi_l|$ has two local maxima at $K_2$ and $K_3$ and one local minimum at $K_1$.

# Appendix B

See table 2.

**Table 2.** Parameter values in figures 16, 15 and 19 for the semi-rational solutions $\Psi_{srp}$ on both constant and periodic backgrounds.

| figure | types of solutions | | parameter values | | | | | |
| | lumps | solitons | fusion case $p_1$ | fusion case $c_1$ | fission case $p_1$ | fission case $c_1$ | periodic background $p_3$ | periodic background $c_3$ |
|---|---|---|---|---|---|---|---|---|
| 16 and 19a | bright–bright | dark–dark | $-1 + i/2$ | $-10 + 10i$ | $1 + i/2$ | $10 + 10i$ | $i$ | $10 + 10i$ |
| 15a and 19b | bright–bright | anti-dark–dark | $-1 + i/2$ | $10i$ | $1 + i/2$ | $10i$ | | |
| 15b and 19c | bright–bright | anti-dark–anti-dark | $-1 + i/2$ | $10 + 10i$ | $1 + i/2$ | $-10 + 10i$ | | |
| 15c and 19d | dark–dark | dark–dark | $-1/2 + i$ | $-10$ | $1/2 + i$ | $10$ | | |
| 15d and 19e | dark–dark | anti-dark–dark | $-1/2 + i$ | $1 + 10i$ | $1/2 + i$ | $1 + 10i$ | | |
| 15e and 19f | dark–dark | anti-dark–anti-dark | $-1/2 + 9i/10$ | $15 + 5i$ | $1/2 + 9i/10$ | $-15 + 5i$ | | |
| 15f and 19g | four-petalled–four-petalled | dark–dark | $-1/2 + i/2$ | $-20 + 10i$ | $1/2 + i/2$ | $20 + 10i$ | | |
| 15g and 19h | four-petalled–four-petalled | anti-dark–dark | $-1/2 + i/2$ | $10i$ | $1/2 + i/2$ | $10i$ | | |
| 15h and 19i | four-petalled–four-petalled | anti-dark–anti-dark | $-1/2 + i/2$ | $15 + 10i$ | $1/2 + i/2$ | $-15 + 10i$ | | |

common parameters: $\kappa = 1$, $a_{21} = b_{11}$, $a_{11} = b_{11} = 0$, $\xi_{i0} = \eta_{i0} = 0$, $i = 1, 2, 3$, $q_1 = p_1^*$, $p_2 = q_2$, and $q_3 = p_3$.

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
