## [Peer Review File · Royal Society Open Science]

Review History

RSOS-201910.R0 (Original submission)

Review form: Reviewer 1

Is the manuscript scientifically sound in its present form?

Yes

Are the interpretations and conclusions justified by the results?

Yes

Is the language acceptable?

Yes

Do you have any ethical concerns with this paper?

No

Have you any concerns about statistical analyses in this paper?

No

Recommendation?

Accept as is

Comments to the Author(s)

This paper is well written, and reveals some interesting nonlinear phenomena. Particularly, the authors have revised the manuscript as the reviewers' suggestion, thus I recommend this paper accepted as this version.

Review form: Reviewer 2**Is the manuscript scientifically sound in its present form?**

Yes

Are the interpretations and conclusions justified by the results?

Yes

Is the language acceptable?

Yes

Do you have any ethical concerns with this paper?

No

Have you any concerns about statistical analyses in this paper?

No

Recommendation?

Accept with minor revision (please list in comments)

Comments to the Author(s)

Constructing various physically meaningful nonlinear wave solutions such as soliton and lump solutions and investigating their underlying dynamics is still one of the active areas of research in the field of integrable systems. In this paper, the authors construct new soliton and (semi-)rational solutions to the y -nonlocal Mel'nikov equation by using the bilinear method and the KP hierarchy reduction method. The contribution of the submitted paper consists in solving two unclear questions proposed on page 3. Two types of solutions with different parameter constraints are investigated. Collision behaviors of the solitons and lumps are analyzed in detail. The results reported in the present paper are interesting and useful as a contribution to better understand the nonlocal integrable systems and may have potential applications in physical. The manuscript is clearly written and logically organized. Hence, I recommend the manuscript for publication in Royal Society Open Science once the authors have completed the following improvements:

1. Equation number is missed for the second equation. Please also check other equations and correct them.
2. It seems that the caption in Figure 2 has missed some description. Please check and correct it.
3. I suggest to adjust the position of some figures to keep the continuity of each paragraph.

Decision letter (RSOS-201910.R0)

Dear Dr Wu

On behalf of the Editors, we are pleased to inform you that your Manuscript RSOS-201910 "General soliton and (semi-)rational solutions of the partial reverse space y -nonlocal Mel'nikov equation with non-zero boundary conditions" has been accepted for publication in Royal Society Open Science subject to minor revision in accordance with the referees' reports. Please find the referees' comments along with any feedback from the Editors below my signature.

Please submit your revised manuscript and required files (see below) no later than 7 days from today's (ie 02-Mar-2021) date. Note: the ScholarOne system will 'lock' if submission of the revision is attempted 7 or more days after the deadline. If you do not think you will be able to meet this deadline please contact the editorial office immediately.

on behalf of Professor Takashi Suzuki (Associate Editor) and Mark Chaplain (Subject Editor)
openscience@royalsociety.org

Associate Editor Comments to Author (Professor Takashi Suzuki):

Comments to the Author:

Your paper is recommended for publication with minor revisions. Take a look at the report and submit the final version.

Reviewer comments to Author:

Reviewer: 1

Comments to the Author(s)

This paper is well written, and reveals some interesting nonlinear

phenomena. Particularly, the authors have revised the manuscript as the reviewers' suggestion, thus I recommend this paper accepted as this version.

Reviewer: 2

Comments to the Author(s)

Constructing various physically meaningful nonlinear wave solutions such as soliton and lump solutions and investigating their underlying dynamics is still one of the active areas of research in the field of integrable systems. In this paper, the authors construct new soliton and (semi-)rational solutions to the y -nonlocal Mel'nikov equation by using the bilinear method and the KP hierarchy reduction method. The contribution of the submitted paper consists in solving two unclear questions proposed on page 3. Two types of solutions with different parameter constraints are investigated. Collision behaviors of the solitons and lumps are analyzed in detail. The results reported in the present paper are interesting and useful as a contribution to better understand the nonlocal integrable systems and may have potential applications in physical. The manuscript is clearly written and logically organized. Hence, I recommend the manuscript for publication in Royal Society Open Science once the authors have completed the following improvements:

1. Equation number is missed for the second equation. Please also check other equations and correct them.
2. It seems that the caption in Figure 2 has missed some description. Please check and correct it.
3. I suggest to adjust the position of some figures to keep the continuity of each paragraph.

===PREPARING YOUR MANUSCRIPT===

===PREPARING YOUR REVISION IN SCHOLARONE===

Author's Response to Decision Letter for (RSOS-201910.R0)

See Appendix A.

Decision letter (RSOS-201910.R1)

Dear Dr Wu,

It is a pleasure to accept your manuscript entitled "General soliton and (semi-)rational solutions of the partial reverse space y -nonlocal Mel'nikov equation with non-zero boundary conditions" in its current form for publication in Royal Society Open Science.

You can expect to receive a proof of your article in the near future. Please contact the editorial office (openscience@royalsociety.org) and the production office (openscience_proofs@royalsociety.org) to let us know if you are likely to be away from e-mail contact – if you are going to be away, please nominate a co-author (if available) to manage the proofing process, and ensure they are copied into your email to the journal.

on behalf of Professor Takashi Suzuki (Associate Editor) and Mark Chaplain (Subject Editor)
openscience@royalsociety.org

Appendix A

Royal Society Open Science RSOS-201910, 'General soliton and (semi-)rational solutions of the partial reverse space y -nonlocal Mel'nikov equation with non-zero boundary conditions', by H. M. Fu, W. S. Lu, J. W. Guo, and C. F. Wu,

Response to comments of the Reviewers

Reviewer comments to Author:

Reviewer: 1

This paper is well written, and reveals some interesting nonlinear phenomena. Particularly, the authors have revised the manuscript as the reviewers' suggestion, thus I recommend this paper accepted as this version.

Reviewer: 2

Constructing various physically meaningful nonlinear wave solutions such as soliton and lump solutions and investigating their underlying dynamics is still one of the active areas of research in the field of integrable systems. In this paper, the authors construct new soliton and (semi-)rational solutions to the y -nonlocal Mel'nikov equation by using the bilinear method and the KP hierarchy reduction method. The contribution of the submitted paper consists in solving two unclear questions proposed on page 3. Two types of solutions with different parameter constraints are investigated. Collision behaviors of the solitons and lumps are analyzed in detail.

The results reported in the present paper are interesting and useful as a contribution to better understand the nonlocal integrable systems and may have potential applications in physical. The manuscript is clearly written and logically organized. Hence, I recommend the manuscript for publication in Royal Society Open Science once the authors have completed the following improvements:

1. Equation number is missed for the second equation. Please also check other equations and correct them.

We have added the number for the second equation.

2. It seems that the caption in Figure 2 has missed some description. Please check and correct it.

We have completed the description of the caption in Figure 2 (see page 11).

3. I suggest to adjust the position of some figures to keep the continuity of each paragraph.

We have adjusted the position of Figs. 11-19 to keep the continuity of each paragraph.